# Nitric-Oxide-Inducing Factors on Vitamin D Changes in Older People Susceptible to Suffer from Sarcopenia

**DOI:** 10.3390/ijerph19105938

**Published:** 2022-05-13

**Authors:** Alfredo Córdova, Alberto Caballero-García, David Noriega-González, Hugo J. Bello, Antoni Pons, Enrique Roche

**Affiliations:** 1Department of Biochemistry, Molecular Biology and Physiology, Faculty of Health Sciences, GIR Physical Exercise and Aging, University of Valladolid, Campus Duques de Soria, 42004 Soria, Spain; 2Department of Anatomy and Radiology, Faculty of Health Sciences, GIR Physical Exercise and Aging, University of Valladolid, Campus Los Pajaritos, 42004 Soria, Spain; alberto.caballero@uva.es; 3Department of Surgery, Ophthalmology, Otorhinolaryngology and Physiotherapy, Faculty of Medicine, Hospital Clínico Universitario de Valladolid, 47005 Valladolid, Spain; davidcesar.noriega@uva.es; 4Department of Mathematics, Escuela de Ingeniería de la Industria Forestal, Agronómica y de la Bioenergía, GIR Physical Exercise and Aging, University of Valladolid, Campus Los Pajaritos, 42004 Soria, Spain; hjbello.wk@gmail.com; 5Research Group on Community Nutrition and Oxidative Stress, University of Balearic Islands, 07122 Palma de Mallorca, Spain; antonipons@uib.es; 6CIBER Fisiopatología de la Obesidad y Nutrición (CIBEROBN), Instituto de Salud Carlos III (ISCIII), 28029 Madrid, Spain; 7Alicante Institute for Health and Biomedical Research (ISABIAL), 03010 Alicante, Spain; 8Department of Applied Biology-Nutrition, Institute of Bioengineering, University Miguel Hernández, 03202 Elche, Spain

**Keywords:** arginine, calcium, citrulline, magnesium, nitrate

## Abstract

Calcium and magnesium, together with vitamin D and the hormones testosterone and cortisol, are key elements in muscle function, to maintain physical fitness. This study aims to analyze if supplementation with NO precursors (L-arginine, L-citrulline and beetroot extract) modulates the circulating levels of calcium, magnesium, vitamin D and steroid hormones in elders. Sixty-one volunteers (65.1 years old, 164.6 cm of height and 71.2 kg of weight) susceptible to develop sarcopenia participated in a physical activity program for 6 weeks. Participants were divided into four groups: one placebo and three taking one of the indicated supplements. Physical capacity was assessed through the following tests: (a) distance covered in 6 min by walking (endurance indicator); (b) hand grip (upper-body strength indicator); (c) time to cover 4 m by walking (speed indicator); and (d) time to perform five full squats (lower-body strength indicator). We concluded that there is a disparity in the association of steroid hormones, vitamin D levels and physical fitness. However, a significant inverse correlation between speed and endurance indicators was observed. Higher circulating vitamin D levels were observed in the L-arginine- and beetroot-supplemented groups. In conclusion, vasodilators increase vitamin D circulating levels that, in the long term, could maintain mineral homeostasis, improving muscular function.

## 1. Introduction

Sarcopenia is a multifactorial clinical disorder associated with aging. There are multiple causes of this pathology, including neurological alterations, endocrine changes and nutritional modifications [1,2]. At the same time, all these changes are related to physical activity [3,4]. In this context, physical fitness is instrumental in the process of muscle regeneration and repair, essential to delay sarcopenia [3]. Insufficient physical activity, low testosterone levels and decreased 25(OH)-vitamin D levels are sarcopenia-developing factors in elders [4]. Studies in mice indicate that 1,25(OH)-vitamin D deficiency may result in the development of sarcopenia through different mechanisms that include oxidative stress activation, senescence of skeletal muscle cells and inhibition of skeletal muscle cell regeneration [5]. In addition, vitamin D deficiency influences the levels of certain minerals [6]. In this context, calcium (Ca), magnesium (Mg) and selenium (Se) seem to be instrumental in preventing and/or treating sarcopenia [7]. Particularly, Ca and Mg play a key role in muscle function and metabolism [8,9,10].

In this context, Ca absorption is dependent on the presence of vitamin D and serum levels of 25(OH)-vitamin D tend to be lower in older adults [5]. This decrease seems to occur with similar dietary intakes of vitamin D when comparing elders with and without sarcopenia [8]. Regarding muscle tissue, altered Ca homeostasis together with oxidative stress in sarcopenia are implicated in mitochondrial disfunction, augmented muscle damage, decreased muscle mass and impaired muscle contractility [11]. Therefore, the restoration of adequate Ca intake delays sarcopenia and osteopenia and reduces the risk of osteoporosis in post-menopausal individuals [7,12]. On the other hand, Mg is involved in protein and ATP synthesis and responsible for muscle relaxation [13]. Chronic Mg deficiency can result in the augmented production of oxygen free radicals and inflammation, being associated with an increased incidence of aging-associated diseases [14]. In this context, healthy elder women daily supplemented with Mg in the form of magnesium oxide (MgO) for 12 weeks seem to improve physical performance, suggesting a role for Mg supplementation in maintaining muscle function and helping to delay sarcopenia [15].

Regarding age-related hormonal changes, prospective cross-sectional and longitudinal studies indicate a progressive decline in serum levels of testosterone. This decrease leads to clinical age-related alterations, such as sarcopenia, low bone mass (osteopenia), anemia and depressive mood [16].

Aging and sarcopenia are associated with an altered biosynthesis of the vasodilator nitric oxide (NO) [17]. Inducible-nitric-oxide synthase (iNOS), responsible for NO synthesis from L-arginine, increases its expression with age and it has been proposed as playing a role in age-related muscle-cell apoptosis. This seems to be mediated by elevated levels of protein damage by nitration with aging, due to an increase in the production of peroxynitrite from NO together with superoxide anion [18,19,20,21]. Supplementation with NO precursors, such as the amino acids L-arginine and L-citrulline or nitrate from beetroot extract, may help to better control NO synthesis [22,23]. These supplements play a key role in muscle function, increasing the bioavailability of nutrients and hormones as a result of their vasodilatory action [24,25]. However, the main part of studies with NO inducers have been performed in the field of sport performance [26,27,28,29], but only few studies have addressed the problem of sarcopenia [25,30].

Taking into account the above information, the present report investigates a population of elders at risk of developing sarcopenia following a physical activity program and supplemented with NO precursors (L-arginine, L-citrulline and beetroot extract). The hypothesis is to study if the supplementation with these NO precursors can modulate the circulating levels of nutrients and hormones that could be instrumental in muscle function. We found that arginine and beetroot extract increase vitamin D levels, a key micronutrient in muscle function and in sarcopenia prevention. No changes in the levels of circulating Ca, Mg and steroid hormones (cortisol and testosterone) involved in the assessment of muscle activity were observed. It seems that longer experiments will be necessary to answer this question.

## 2. Materials and Methods

### 2.1. Participants

In this study, 61 subjects (60–73 years old) participated (Table 1). The participants did so voluntarily and signed the informed consent form. Previous to being conducted, the study was explained in all its terms (objectives, technical staff involved and methodology). All participants underwent a complete medical examination in collaboration with their family doctor. In order to be admitted in the study, inclusion and exclusion criteria were established according to the suggestions and comments of the family doctor. Participants were divided into 4 groups: placebo group (PL), L-citrulline-malate-supplemented group (CM), L-arginine-supplemented group (ARG) and nitrate (beetroot extract)-supplemented group (BEET). All subjects participated regularly in a physical activity program. None of the volunteers dropped out of the study after 6 weeks of intervention.

The study was approved by the Ethics Committee of the Hospital General de Alicante (Ref. CEIm PI2019-070). The inclusion criteria were: being over 60 years old and in good health (not affected by any exclusion criteria). Exclusion criteria were: (a) history of dementia; (b) moderate/severe chronic obstructive pulmonary disease (COPD) with Bodex index C or D; (c) functional limitation according to the Barthel scale (less than 100 = maximum score) and the Lawton–Brody scale (less than 8 = maximum value); (d) recent acute myocardial infarction (3–6 months) or unstable angina; (e) uncontrolled atrial or ventricular arrhythmias, dissecting aortic aneurysm, severe aortic stenosis and acute endocarditis/pericarditis; (f) uncontrolled hypertonia (>180/100 mmHg); (g) acute thromboembolic disease; (h) acute/chronic heart failure with NYHA (New York Heart Association) scale > II; (i) acute/chronic respiratory failure; (j) uncontrolled orthostatic hypotension; (k) diabetes mellitus with acute decompensation or uncontrolled hypoglycaemia; (l) recent bone fracture (within the last month); (m) previous supplementation with amino acids or other nutritional compounds to enhance physical performance; (n) any other circumstances deemed by their physician to preclude physical activity. At the beginning and at the end of the study, after the 6th week of the nutritional intervention, blood tests and physical assessment tests were performed.

The physical activity program consisted in a warm-up for 10 min, balance exercises for 5 min, aerobic endurance (walking/slow running) for 10 min and aerobic resistance (overload exercises with balls, dumbbells, rubber bands, etc.) for 20 min.

### 2.2. Supplementation

The study was a double-blind, placebo-controlled trial (Figure 1). The dose of the supplements were 3 g/day for CM, 2 g/day for ARG, 300 mg/day of beetroot extract rich in nitrate for BEET and 3 g of lactose and starch for PL group. All supplements were administered orally in capsules. The placebo was prepared with lactose and starch in capsules of the same size, weight and color as the capsules used for the experimental groups. Each week, a member of the research team (physician) was responsible for the weekly distribution of the capsules and the recording of any clinical observations that might occur. The doctor in charge of distribution was unaware of the composition of the capsules during the entire intervention period. The capsules were blinded by codes corresponding to placebo or supplement. All groups (PL, CM, ARG and BEET) took the corresponding capsules during the 6-week follow-up period.

### 2.3. Physical Testing

The assessment of physical capacities was performed as follows: (a) A 6 min test (endurance) was performed on an approved 400 m athletics track and the distance covered after 6 min was measured; (b) Hand grip strength using a JAMAR digital dynamometer (0–90 kg) (Performance Health, Warrenville, IL, USA); (c) Gait speed: the time to walk a distance of 4 m was measured by installing photoelectric cells at both ends. To minimize variability, volunteers were asked to start walking 5 m before the timed zone at their usual walking speed; (d) Squat test to determine the time taken to perform 5 full squats from a seated position, without arm assistance, at the fastest possible speed. Endurance was performed only once in the athletics track. Hand grip, gait speed and squat tests were performed 3 times and the mean was calculated.

For the monitoring of the physical activity, a record was made at each of the training sessions. One of the members of the research team participated and collected the required information at the end of each session.

### 2.4. Blood Analysis

The first blood extraction was performed on the first day that participants started the study and the second one was performed on the last day of the intervention, after 6 weeks. Blood samples were taken at 08:00 a.m. with participants fasting and after 8–12 h of correct resting the previous night. Serum (10 mL) was collected in Vacutainer tubes with a clot activator. In addition, 3–5 mL of blood was placed in EDTA tubes to obtain plasma. Blood extractions were performed while the participants remained seated. Plasma tubes were stored in refrigerated containers at a temperature of 4 °C and serum tubes were frozen at −20 °C.

Circulating parameters included: glucose, urea, uric acid, total cholesterol, LDL-cholesterol, HDL-cholesterol, triglycerides and total protein were determined with the Architect ci8200^®^ analyzer (Abbott, Madrid, Spain). Ca and Mg were determined by atomic absorption spectrophotometry (PerkinElmer Inc., Waltham, MA, USA). Testosterone, cortisol and vitamin D were analyzed on an Architect 2000^®^ system. The sensitivity of each determination is indicated in the instruction manual of the different apparatus used. Blood pressure was performed 3 times and the mean was calculated.

### 2.5. Statistics

Statistical analyses were performed with the IBM statistical package SPSS version 24. Data were expressed as a mean ± standard error of the mean (X ± SEM). Afterwards, we tested for the normality of distribution using the Shapiro-Wilk and Kolmogorov-Smirnov tests. Levene’s homogeneity of variances was studied. Frequencies were estimated for categorical variables and means of central tendency and standard deviation were calculated for quantitative variables. The effect of NO precursor supplementation on changes in circulating values of the analyzed parameters was tested by a two-way ANOVA test with supplementation (PL, CM, ARG and BEET) and 6 weeks of training (baseline vs. end of intervention) as factors, also performing a post hoc test (Bonferroni and Scheffé) or one-way ANOVA on the variables that showed a level of statistical significance. In all calculations, the level of statistical significance was *p* < 0.05. A Pearson correlation test was performed between circulating parameters and fitness parameters.

## 3. Results

Anthropometric parameters, blood pressure values, levels of circulating calcium, magnesium, vitamin D, steroid hormones and the results of the physical condition tests of the men and women that participated in the intervention are shown in Table 1.

Compared to the women, the men that participated in the study were significantly taller and displayed a significant higher body weight and body mass index, significant lower body fat percentage, lower circulating Mg and cortisol levels and higher circulating testosterone levels. Circulating vitamin D and Ca values were very similar in both genders. Regarding physical capacities, men showed a significantly higher strength and endurance than the women, but a similar walking speed to them. Their time to perform the five squats was shorter or, in other words, the men were faster to perform the squats. Finally, women had a significantly lower testosterone/cortisol (T/C) ratio than men.

Regarding changes in the other circulating parameters during the intervention, glucose levels, lipid profile and markers of protein metabolism were within the healthy range (Table 2). No significant effect of supplementation was observed.

Correlations between the circulating levels of Ca, Mg, vitamin D, testosterone, cortisol and fitness indicators were analyzed in the men and women that participated in the study (Table 3). The men showed significant inverse correlations between vitamin D and Mg levels, thus high levels of vitamin D correlated with low levels of circulating Mg (intracellular mineral). The men also presented significant inverse correlations between the time to walk 4 m (speed indicator) and meters walked in 6 min (endurance indicator), thus the fastest men were also the most enduring. Similarly, the men showed significant direct correlations between circulating cortisol and testosterone levels and the T/C index. The T/C ratio was correlated positively with the time that the men took to walk 4 m, thus high T/C index values correlated with a slow walking speed. Hand grip strength levels in the men were correlated significantly and directly with cortisol levels and inversely with vitamin D levels. Endurance in the men was correlated positively with strength and walking speed, thus high strength was associated with more meters walked in 6 min and a shorter time to walk 4 m. Likewise, the men taking longer to walk 4 m also took longer to perform the five squats. For women, the situation was similar, although there were some particularities. Circulating Mg levels were significantly and positively correlated with Ca levels. Cortisol levels in the women significantly correlated with testosterone levels, similarly to the men. Endurance (meters walked in 6 min) in the women was correlated positively with circulating Mg levels and inversely with circulating testosterone levels. As in the case of the men, high walking speed was associated with more distance walked in the endurance test. Similarly, the women that took longer to walk 4 m also took longer to perform the five squats. However, unlike the men, the women that walked a longer distance in the endurance test also took less time to perform the five squats.

The supplementation with the L-citrulline or L-arginine or nitrate-rich beetroot extracts for 6 weeks did not significantly influence the circulating Ca and Mg levels. However, the supplementation influenced the circulating vitamin D levels, depending on the type of supplement ingested (Table 4). The supplementation for 6 weeks with the L-arginine or beetroot extracts significantly increased the circulating vitamin D levels, while the placebo and L-citrulline-supplemented groups maintained their initial levels. In addition, after 6 weeks of supplementation, the circulating vitamin D levels were significantly higher in the L-arginine- or beetroot-extract-supplemented groups than in the placebo or L-citrulline-supplemented groups.

Finally, Table 5 shows the changes in testosterone, cortisol and T/C ratio at the beginning and at the end of the 6 weeks of supplementation with placebo or NO precursors. No significant effects of the supplementation were observed, except for a significant effect on the testosterone levels and T/C ratio in the group supplemented with the beetroot extract. Nevertheless, these circulating values were high at the beginning of the intervention in this group; thus, changes do not seem attributable to beetroot extract consumption.

## 4. Discussion

Optimal fitness through regular physical activity performance is one of the actions that delay the onset of sarcopenia [31]. Maintaining physical activity during ageing is instrumental for skeletal muscle function and response to the anabolic effects of nutrition [31]. Age-related muscle functional decline is different when comparing men and women, mainly after 60 years of age [32]. In this context, the men and women that participated in the present study were physically fit but at risk of sarcopenia due to their age range. The cut-off points for hand strength in the diagnosis of sarcopenia are around 16 kg for women and 27 kg for men [33], well below the values of the study participants. In addition, the physical tests performed to establish the fitness level are correlated between them, allowing to define an optimal general fitness level for participants. Nevertheless, the decline with age in fitness is similar in men and women, although slight differences have been reported depending on the type of strength determined [31]. Isometric knee extension and knee flexion strength experience similar rapid declines in both genders by the end of the sixth decade of life. However, there is an abrupt age-related decline in the maximal knee extension torque that appears earlier in women (age 41.8 years) than in men (age 66.7 years). Nevertheless, the loss of maximal knee flexion torque strength occurs faster with age in men (age 49.3 years); meanwhile, it is not observed in women. This information was important to design the training program in the present report. Nevertheless, the gender differences observed in the physical test correlations could be associated with the differential rate of age-related functional decline mentioned before [31].

The metabolism activation necessary for physical performance requires adequate levels of testosterone. In this context, low testosterone circulating levels in men are considered to be risk factors for the onset of sarcopenia [4]. The association of testosterone with sarcopenia has been well established in men, but not fully understood in women. Low levels of free testosterone may be a significant determinant of decreased muscle strength in women and physical performance in men. However, they do not predict loss of muscle mass in the short term in both men and women [34]. In addition, clinical trials indicate that exogenous testosterone improves cognitive and skeletal muscle functions in post-menopausal women [35]. In the present study, circulating testosterone levels were correlated positively with cortisol levels and the T/C ratio, while a poor correlation was observed with the fitness parameters in both men and women. Only a negative correlation was noticed between testosterone levels and endurance capacity in women, and a positive correlation between cortisol levels and strength and between the T/C ratio and speed in men, while there was no correlation in women. The T/C ratio could be representative of the degree of balance between anabolic and catabolic processes; thus, a lower ratio is indicative of the predominance of catabolic processes [28,29]. In this sense, the obtained results suggest that the women that participated in this study have a higher catabolic profile than the men at this age segment. Nevertheless, the testosterone, cortisol, Mg and Ca levels of the men and women that participated in the study were within ranges considered healthy [7,8,9,10]. Nevertheless, the evidence from the present report seems to suggest that testosterone and cortisol levels as well as the T/C ratio appear to be poor fitness indicators in men over 60 years of age, while in women, no correlation was found. In the same line, a previous study found a disparity in the association between circulating levels of free testosterone and frailty in older men and women [36]. Altogether, all these results suggest that different biological mechanisms may be involved, such as increased oxidative stress, delayed protein synthesis and low tissue repair efficiency. Additional research will be necessary to decipher these questions.

As mentioned before, a low circulating level of vitamin D is a risk factor to develop sarcopenia [4]. In this context, the circulating levels of vitamin D were correlated negatively with strength in the men, whereas in the women, no correlation was observed. To prevent bone health problems in elders, the US Institute of Medicine recommends vitamin D supplementation with 600 for the population aged 1–70 years and 800 IU/day for the population aged of 71 years and older, with a maximum of 4000 IU/day to maintain circulating 25(OH)-vitamin D levels above 20 ng/mL [37]. On the other hand, the US Endocrine Society recommends maintaining circulating levels of vitamin D above 30 ng/mL through the intake of 1500–2000 IU daily [38]. In the present report, vitamin D levels seem to be within the adequate ranges, making vitamin D supplementation unnecessary. However, in a previous study by our group carried out in elders (60–62 years), we observed that vitamin D supplementation (2000 IU/day) resulted in a decrease in muscle damage indicators and with circulating 25(OH)-vitamin D levels of 31.3 ng/mL [39]. Therefore, vitamin D is key in the maintenance of skeletal muscle homeostasis. A direct association of vitamin D levels with power, strength, physical performance and reduced falls has been documented [40,41,42,43].

The adequate levels of 25(OH)-vitamin D of the present study could favor correct circulating levels for Ca and Mg, resulting in a correct phosphocalcic metabolism. In this line, Mg is incorporated into the bone matrix and contributes to bone structure. In mice, it has been shown that Mg deficiency decreases osteoprotegerin and increases the receptor activation of the nuclear factor-κB ligand (RANKL), contributing to increased osteoclastogenesis [44]. In the present study, circulating Mg levels were within the healthy range in both the men and women. In this context, salivary Mg was significantly higher in the women than in the men, while urine Mg was significantly higher in the men than in the women [45]. This observation could explain why the women had significantly higher serum Mg levels than the men in the present report. Adequate vitamin D levels are in line with optimal serum levels of Ca. This correlation might be interpreted as adequate levels of these micronutrients are key to maintain phosphocalcic metabolism. A predominant role is played by vitamin D performing the tight regulation of Mg and Ca levels in the serum and bone stores as well as in excretion through urine or the gastrointestinal tract. What is agreed by experts and most of the scientific societies involved is to consider vitamin D deficiency at values below 20 ng/mL [46]. Adequate levels above this figure enable correct bone homeostasis for Ca and Mg. In this line, some authors have found a lower Mg content and larger hydroxyapatite crystals in the bones of women with osteoporosis [47]. On the other hand, inadequate serum Mg concentrations result in low serum Ca concentrations, resistance to the action of the parathyroid hormone (PTH) and resistance to some of the effects of vitamin D (calcitriol), resulting in increased bone loss [48,49]. Nevertheless, the men and women that participated in this study were not in a state of vitamin D deficiency. Furthermore, circulating Mg levels positively correlated with Ca levels and endurance capacity in the women, while the same parameters were negatively correlated with vitamin D in the men. These correlations suggest a role for Mg in preventing or delaying age-related decline in physical performance in women. This correlation strengthens the concept that daily supplementation with MgO for 12 weeks seems to improve physical performance in healthy elderly women [15].

Finally, supplementation with NO precursors, such as L-citrulline, L-arginine or beetroot extracts, favors vasodilation, allowing a high bioavailability of nutrients and hormones to muscles and thus helping in physical performance. However, the supplementation did not alter the basal circulating levels of Ca, Mg, testosterone and cortisol. Nevertheless, there was a significant increase in circulating vitamin D levels after 6 weeks of L-arginine- and beetroot-extract supplementation compared to L-citrulline or placebo. Likely, a longer intervention should be necessary to observe increased circulating levels of minerals and hormones. In this line, other beneficial actions of vitamin D cannot be ignored: the potentiation of vascular tone by increasing endothelial NO production and reduced oxidative stress and inflammation [50,51,52,53]. Additional research will be necessary to test all these hypotheses. In any case, this study has some limitations: (A) Supplements together with the physical activity program need to be tested in people with sarcopenia; (B) A longer intervention needs to be performed to verify changes in circulating mineral levels; (C) The role of other nutrients in the prevention of sarcopenia needs to be addressed; and (D) The design of a more adequate and adapted physical activity planning is necessary. Altogether, the main finding of this study is that supplementation with certain precursors of NO (Arg and nitrites from beetroot extracts) increases the circulating levels of vitamin D, a key micronutrient in muscle metabolism and thereby in physical activity performance and sarcopenia prevention. All these factors are key to prevent the onset of sarcopenia.

## 5. Conclusions

In conclusion, supplementation with the vasodilators L-arginine and nitrate-rich beetroot extracts allow the increase in circulating vitamin D levels in men and women over 60 years of age at risk of undergoing sarcopenia and following a physical activity program. Nevertheless, further research likely with longer interventions has to elucidate if this increase can favor the mineral homeostasis, particularly for calcium and magnesium, that is key in muscle function and bone metabolism (preventing osteosarcopenia). Regarding hormones related to physical fitness, there is a disparity in the association of testosterone, cortisol and vitamin D levels in the same group of elders.

## Figures and Tables

**Figure 1 ijerph-19-05938-f001:**
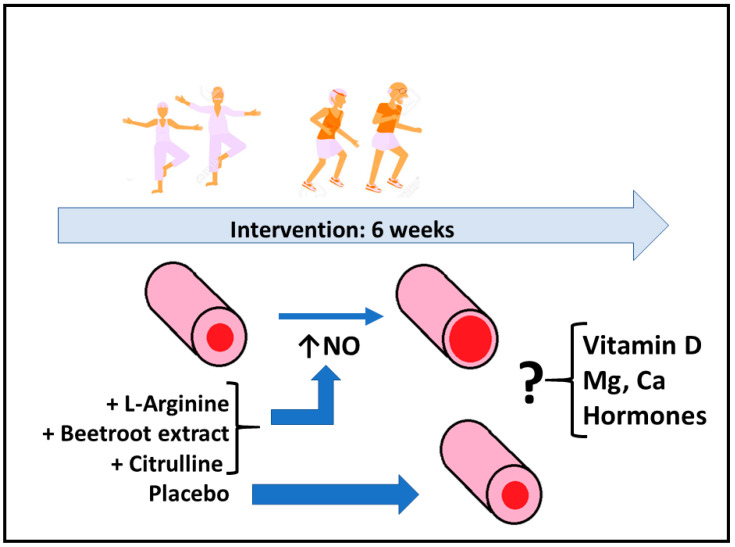
Scheme of the study hypothesis (see text for more details).

**Table 1 ijerph-19-05938-t001:** Main characteristics of participants at the beginning of the study.

Parameter	Men	Women	ANOVA *p*
*n*	29	32	
**Anthropometric parameters**			
Height (cm)	171.0 ± 4.3	158.1 ± 6.1 *	* 0.000
Age (years)	64.8 ± 3.6	65.4 ± 4.4	0.781
Weight (kg)	78.4 ± 9.0	63.9 ± 7.9 *	* 0.000
BMI (kg/m^2^)	26.6 ± 2.3	24.0 ± 2.9 *	* 0.032
Fat (%)	24.4 ± 2.7	34.9 ± 5.6 *	* 0.000
**Blood pressure values**			
Systolic (mmHg)	133 ± 17	130 ± 21	0.070
Diastolic (mmHg)	79 ± 10	77 ± 11	0.338
**Circulating parameters**			
Calcium (mg/dL)	9.5 ± 0.2	9.6 ± 0.3	0.097
Magnesium (mg/dL)	2.1 ± 0.1	2.2 ± 0.1 *	* 0.003
Vitamin D (ng/mL)	25.9 ± 8.0	26.1 ± 9.3	0.892
Testosterone (ng/mL)	4.2 ± 1.2	0.4 ± 0.2 *	0.000
Cortisol (μg/mL)	14.6 ± 3.2	17.0 ± 3.6 *	* 0.001
Testosterone/Cortisol Index	29.4 ± 9.2	2.3 ± 0.9 *	* 0.000
**Physical condition tests**			
Strength Dynamometry (kg)	42.1 ± 8.6	28.5 ± 7.8 *	* 0.000
Endurance 6 min (m)	1055 ± 220	820 ± 143 *	* 0.000
4 m walking speed (s)	2.2 ± 0.3	2.3 ± 0.3	0.122
5 Squats (s)	10.2 ± 2.0	11.3 ± 1.9 *	* 0.011

* The difference between women and men is significant at *p* < 0.05 by one-way ANOVA test. Abbreviations: BMI, body mass index.

**Table 2 ijerph-19-05938-t002:** Additional circulating parameters.

Group:	PL		CM		ARG		BEET	
Parameter	T1	T2	T1	T2	T1	T2	T1	T2
Glucose (mg/dL)	89.7 ± 6.2	90.0 ± 6.0	90.0 ± 8.0	83.7 ± 6.2	88.2 ± 7.7	81.5 ± 11.5	90.2 ± 2.7	86.5 ± 4.5
Urea (mg/dL)	42.2 ± 9.2	40.5 ± 6.5	40.2 ± 2.7	40.5 ± 3.5	43.5 ± 11.5	38.0 ± 7.0	36.7 ± 5.2	37.0 ± 6.0
Uric acid (mg/dL)	5.15 ± 0.9	5.0 ± 0.7	4.8 ± 0.7	4.8 ± 0.7	4.7 ± 0.7	4.7 ± 0.6	4.4 ± 0.7	4.8 ± 1.0
Cholesterol (mg/dL)	212 ± 18.5	236 ± 22.5	220 ± 17.5	212 ± 18.5	242 ± 17.0	239 ± 30.0	211 ± 8.5	214 ± 10.5
HDL (mg/dL)	66.0 ± 7.0	62.0 ± 6.0	61.0 ± 3.5	59.0 ± 5.0	64.0 ± 8.0	59.0 ± 6.0	58.7 ± 5.2	59.5 ± 6.5
Triglycerides (mg/dL)	81.2 ± 7.2	89.2 ± 19.3	78.7 ± 15.7	79.2 ± 16.7	80.5 ± 17.5	73.5 ± 13.5	90.0 ± 24.0	77.5 ± 14.5
Total proteins (g/dL)	7.2 ± 0.3	7.3 ± 0.2	6.9 ± 0.1	7.1 ± 0.1	7.1 ± 0.2	7.0 ± 0.2	7.3 ± 0.1	7.0 ± 0.3

Groups: placebo (PL) and supplemented with citrulline malate (CM), arginine (ARG) and beetroot extract (BEET).

**Table 3 ijerph-19-05938-t003:** Correlations between physical performance and calcium, magnesium and steroid hormones in the women and men that participated in the study.

	WOMEN	MEN
Calcium (Ca)	Mg (0.269) *p* = 0.05 * T/C index (−0.278) *p* = 0.046 *	
Magnesium (Mg)	Ca (0.269) *p* = 0.05 * Endurance (m) (0.271) *p* = 0.052 *	Vitamin D (−0.330) *p* = 0.038 *
Vitamin D		Mg (−0.330) *p* = 0.038 * Strength (kg) (−0.364) *p* = 0.024 *
Testosterone (T)	C (0.440) *p* = 0.001 ** T/C index (0.717) *p* = 0.000 ** Endurance (m) (−0.289) *p* = 0.038 *	C (0.396) *p* = 0.011 * T/C index (0.766) *p* = 0.000 **
Cortisol ©	T (0.440) *p* = 0.001 **	T (0.396) *p* = 0.011 * Strength (kg) (0.532) *p* = 0.001 **
T/C index	T (0.717) *p* = 0.000 ** Ca (−0.278) *p* = 0.046 *	T (0.766) *p* = 0.000 ** Speed (s) (0.379) *p* = 0.019 *
Strength (kg)		C (0.532) *p* = 0.001 ** Vitamin D (−0.369) *p* = 0.024 * Endurance (m) (0.332) *p* = 0.036 *
Endurance (m)	Mg (0.271) *p* = 0.05 * Speed (s) (−0.634) *p* = 0.000 ** Squats (s) (−0.457) *p* = 0.001 **	Strength (kg) (0.437) *p* = 0.006 ** Speed (s) (−0.371) *p* = 0.022 *
Speed (s)	Endurance (m) (−0.634) *p* = 0.000 ** Squats (s) (0.544) *p* = 0.000 **	Index T/C (0.379) *p* = 0.019 * Endurance (m) (−0.371) *p* = 0.022 * Squats (s) (0.486) *p* = 0.002 **

Pearson’s correlation and *p*-values. * Correlation is significant at a *p* < 0.05 level (bilateral). ** Correlation is significant at a *p* < 0.01 level (bilateral).

**Table 4 ijerph-19-05938-t004:** Effects of supplementation with L-citrulline or L-arginine or nitrate-rich beetroot extracts on the calcium, magnesium and vitamin D plasma levels.

Parameter	Groups	T1	T2	ANOVA
Ca (mg/dL)	PL	9.6 ± 0.3	9.6 ± 0.3	S (0.719) T (0.309) SxT (0.246)
	CM	9.5 ± 0.2	9.6 ± 0.3	
	ARG	9.6 ± 0.2	9.5 ± 0.2	
	BEET	9.7 ± 0.4	9.5 ± 0.3	
Mg (mg/dL)	PL	2.2 ± 0.1	2.1 ± 0.1	S (0.496) T (0.211) SxT (0.604)
	CM	2.2 ± 0.2	2.1 ± 0.1	
	ARG	2.2 ± 0.1	2.2 ± 0.2	
	BEET	2.1 ± 0.1	2.1 ± 0.2	
Vitamin D (ng/mL)	PL	22.6 ± 7.3	22.7 ± 6.4	S (0.0.032) T (0.05) SxT (0.553)
	CM	24.7 ± 10.3	26.5 ± 9.1	
	ARG	26.6 ± 6.6	32.7 ± 9.5 ^a,b^	
	BEET	26.0 ± 7.4	32.5 ± 7.0 ^a,b^	

Groups: Placebo (PL) and supplemented with citrulline malate (CM), arginine (ARG) and beetroot extracts (BEET). Abbreviations: T1, beginning of intervention; T2, end of intervention. S indicates the effect of the supplement (*p*-value); T indicates the effect of the intervention (*p*-value); and SxT indicates the interaction of the supplement and intervention effects (*p*-value) by a two-way ANOVA analysis. ^a^ Significant difference (*p* < 0.05) comparing T1 vs. T2. ^b^ Significant difference (*p* < 0.05) comparing ARG or BEET vs. PL or CM by a one-way ANOVA analysis.

**Table 5 ijerph-19-05938-t005:** Testosterone and cortisol levels at the beginning (T1) and end (T2) of the intervention with NO precursors for 6 weeks.

Parameter	Group	T1	T2	ANOVA
Testosterone (T) (ng/mL)	PL	1.7 ± 1.9	1.6 ± 1.9	S (0.031) T (0.671) SxT (0.998)
	CM	1.9 ± 1.9	1.7 ± 1.8	
	ARG	2.1 ± 2.3	1.9 ± 2.3	
	BEET	3.7 ± 2.5 *	3.4 ± 2.2	
Cortis © (C) (µg/mL)	PL	16.6 ± 4.5	16.8 ± 3.9	S (0.561) T (0.123) SxT (0.387)
	CM	15.9 ± 4.1	15.9 ± 2.8	
	ARG	17.6 ± 3.4	14.5 ± 2.7	
	BEET	16.1 ± 3.5	13.8 ± 3.1	
T/C index	PL	11.1 ± 13.5	9.66 ± 12.4	S (0.007) T (0.955) SxT (0.972)
	CM	13.1 ± 13.4	11.2 ± 12.0	
	ARG	12.7 ± 14.3	14.0 ± 16.9	
	BEET	25.7 ± 18.6 *	27.0 ± 18.1 *	

Groups: Placebo (PL) and supplemented with citrulline malate (CM), arginine (ARG) and beetroot extracts (BEET). S indicates the effect of the supplement (*p*-value); T indicates the effect of the intervention (*p*-value); and SxT indicates the interaction of the supplement and intervention (*p*-value) by a two-way ANOVA analysis. * Significant difference (*p* < 0.05) of the BEET group with respect to the PL group by a one-way ANOVA analysis.

## Data Availability

Data supporting reported results are available upon reasonable request from A.C.-M.

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
