# Peer review of "Nitric-Oxide-Inducing Factors on Vitamin D Changes in Older People Susceptible to Suffer from Sarcopenia"

_ijerph, 2022, doi:10.3390/ijerph19105938_

Round 1

Reviewer 1 Report

The introduction sections needs a little work to tie together why you choose the biochemical and vitamin/mineral status to measure as biomarkers for the intervention. The rationale as to why the supplements you tested may have any impact on serum vitamin/mineral status is not clear.

Section 4.2, I believe you meant "lactose" and not "lacose", please double check and correct.

The conclusions section is too brief. This section actually does not summarize the findings (needs to) but rather focuses on physical fitness relationship with vitamins/minerals and hormones. Your point of the study in part was to see if any of the supossed vasodialators you tested would also have concomitant effects or impacts on serum or circulating levels of what was evaluated. This is not at all covered in the conclusions sections, please edit and update.

Author Response

REVIEWER-1

The introduction sections need a little work to tie together why you choose the biochemical and vitamin/mineral status to measure as biomarkers for the intervention. The rationale as to why the supplements you tested may have any impact on serum vitamin/mineral status is not clear.

ANSWER: We appreciate the comment of the Reviewer. We think that the Introduction needs some order in the presentation of the main ideas that we develop in the manuscript. We have changed the structure of Introduction as follows:

Lines 42-54: We have introduced sarcopenia developing factors, indicating that one of these factors are low circulating levels of vitamin D. This vitamin is instrumental in mineral homeostasis, particularly in calcium and magnesium.

Lines 55-68: We presented the role of Ca and Mg in muscle function and thereby in the development of sarcopenia when levels of these minerals are low.

Lines 69-72: Muscle activity depends as well from the activity of specific hormones which concentration declines in sarcopenia, leading a less active muscle tissue.

Lines 73-84: Finally, we start to introduce our rational. We indicated that the vasodilator NO increases with aging leading to oxidative stress. However, a controlled production of NO with supplements may result in an increase of blood flow to muscle and thereby in more bioavailability of vitamin D. As a result, circulating levels of minerals (Ca and Mg) and hormones increase as well, being instrumental for muscle activity and function, delaying sarcopenia.

Lines 85-94: Presentation of the working hypothesis.

Section 4.2, I believe you meant "lactose" and not "lacose", please double check and correct.

ANSWER: Typo has been corrected (see line 315).

The conclusions section is too brief. This section actually does not summarize the findings (needs to) but rather focuses on physical fitness relationship with vitamins/minerals and hormones. Your point of the study in part was to see if any of the supposed vasodilators you tested would also have concomitant effects or impacts on serum or circulating levels of what was evaluated. This is not at all covered in the conclusions sections, please edit and update.

ANSWER: Conclusion section has been changed accordingly (see lines 370-377).

Reviewer 2 Report

This is an interesting study. However, according to the findings, perhaps it had not been adequately planned and executed.

Usually, interventions in the elderly may take longer than six weeks to appear. Therefore, the authors are correct in stating that a more extended follow-up period is required to notice any effects of the supplements tested on the physical performance of the elderly.

Also, the testosterone, cortisol, Mg, Ca, and vitamin D levels of the men and women participating in the study being within healthy ranges may be a problem since the authors may have targeted a subgroup of elderly participants not in need of correcting those variables.

Also, a previous study discussed by the authors presented a disparity in the association between circulating levels of free testosterone and frailty in older men and women. Indeed different biological mechanisms could be involved, that the authors should consider.

Why are the Materials & Methods part before the Conclusions? Please amend.   

Author Response

REVIEWER-2

This is an interesting study. However, according to the findings, perhaps it had not been adequately planned and executed.

Usually, interventions in the elderly may take longer than six weeks to appear. Therefore, the authors are correct in stating that a more extended follow-up period is required to notice any effects of the supplements tested on the physical performance of the elderly.

ANSWER: We appreciate the comment of the Reviewer and he has right indicating that longer intervention studies are necessary. We have indicated this point in the new Conclusion section. Nevertheless, the main idea of the intervention was to study early events in the prevention of sarcopenia. In this line, we have found that some vasodilators can quickly increase vitamin D levels. Taking into account the low levels of this vitamin in this age range, this result could be a first step in the prevention of sarcopenia and even other age-related diseases. This point is indicated as well in the new Conclusion section. See lines 370-377.

Also, the testosterone, cortisol, Mg, Ca, and vitamin D levels of the men and women participating in the study being within healthy ranges may be a problem since the authors may have targeted a subgroup of elderly participants not in need of correcting those variables.

ANSWER: Reviewer is right that the studied population is a healthy group of elders. However, as we mentioned in the previous answer, we wanted to explore prevention strategies in sarcopenia. Usually, elder people go to the physician when the disease is declared, otherwise said, when it is too late and the cost for the national health systems is significant. The supplements used are not expensive and could work extending the quality of life of these persons.

Also, a previous study discussed by the authors presented a disparity in the association between circulating levels of free testosterone and frailty in older men and women. Indeed different biological mechanisms could be involved, that the authors should consider.

ANSWER: Our results and the results obtained in Reference 35, strongly suggest that testosterone and cortisol levels as well as T/C ratio seem to be poor fitness indicators in elders compared to young active people. We agree with the Reviewer that different mechanisms seem to operate such as increased oxidative stress, delayed protein synthesis, low tissue repair efficiency, among others. The proposed mechanisms need further research (see lines 214-217).

Why are the Materials & Methods part before the Conclusions? Please amend.

ANSWER: We are following Journal Instructions.

Reviewer 3 Report

  • The authors studied nitric oxide inducing factors on serum minerals changes in older people susceptible to suffer sarcopenia. The study is very informative and adds value to the current research body; however, some points should be addressed.
  • The English editing is needed as there are several errors that I mentioned some of them in the following:

Abstract

  • The abstract needs to contain anthropometric characteristics of participants such as age, height, weight, etc.
  • Lines 29: Correct the word “indcated” to “indicated”.
  • Lines 34: Correct the word “oberved” to “observed”.
  • The p-value should be mentioned for all the significant changes mentioned in the abstract.
  • Keep the keywords between 3-5 words.

Introduction

  • Lines 43-44: Provide at least two references to support the following sentence:

“At the same time, all these changes are related to physical activity.”

  • Line 45: Change “low physical activity” to “insufficient physical activity”.
  • Lines 45-47: Rewrite the following sentence:

“Low physical activity, smoking, leanness, low testosterone levels and decreased 25(OH)-vitamin D levels are sarcopenia developing factors in elders [4].”

  • Provide a reference(s) to support the following sentence.

“Therefore, vitamin D deficiency influences the levels of certain minerals,”

  • Lines 64-66: Rewrite the following sentence:

“Healthy elder women daily supplemented with magnesium oxide (MgO) for 12 weeks seem to improve physical performance, suggesting a role for Mg supplementation in preventing or delaying age-related decline in physical fitness [16].”

  • Lines 72-74: Provide at least two references to support the following sentence:

“Supplementation with NO precursors such as the amino acids L-arginine and L-citrulline or nitrate from beetroot extract, may help to control NO synthesis.”

  • The study hypothesis needs to be added at the end of “Introduction”.

Methods

  • “Materials and Methods” should be after the “Introduction” section.
  • Lines 261-262: What was the physical activity program? It needs to be explained.
  • Day-to-day test reliability, CV range, and intraclass correlation coefficients for the assessments need to be included for ALL the assessments.
  • Add a schematic representation of the study procedures to the “Methods” section.

Results

  • “Results” should be after the “Materials and Methods” section.

Discussion

  • “Discussion” should be after the “Results” section.
  • The discussion section of this manuscript is the biggest concern. It needs to be elaborated and described more comprehensively to prepare for publication.
  • What is the novelty of this study? It needs to be mentioned in the “Discussion”.
  • The Study limitations part is completely missing from the manuscript, and there are numerous limitations to be listed.

Author Response

REVIEWER-3

  • The authors studied nitric oxide inducing factors on serum minerals changes in older people susceptible to suffer sarcopenia. The study is very informative and adds value to the current research body; however, some points should be addressed.

ANSWER: We thank the Reviewer for this comment. However, supplements did not significantly influence circulating Ca and Mg levels. Supplements (Arg and beetroot) only influence vitamin D levels. This point has been indicated in the title, giving a more precise information to the reader.

  • The English editing is needed as there are several errors that I mentioned some of them in the following:

ANSWER: If the manuscript is accepted, we will ask English editing services to the Editorial before publications. In any case, we have corrected the typos indicated.

Abstract

  • The abstract needs to contain anthropometric characteristics of participants such as age, height, weight, etc.

ANSWER: The Abstract has a limit of 200 words (see instructions for Authors). However, we could include general anthropometric characteristics without surpassing this limit (see line 28).

  • Lines 29: Correct the word “indcated” to “indicated”.

ANSWER: Typo has been corrected (see line 30).

  • Lines 34: Correct the word “oberved” to “observed”.

ANSWER: Typo has been corrected (see line 35).

  • The p-value should be mentioned for all the significant changes mentioned in the abstract.

ANSWER: If we indicate the p values in the Abstract, we surpass the word limit indicated in the Instructions for Authors. We guess that readers can have access to this information in the corresponding tables.

  • Keep the keywords between 3-5 words.

ANSWER: Instructions for Authors indicate between 3-10 keywords. Nevertheless, we have adjusted to 5 keywords (see line 39).

Introduction

  • Lines 43-44: Provide at least two references to support the following sentence:

“At the same time, all these changes are related to physical activity.”

ANSWER: References 3 and 4 have been cited (line 45).

  • Line 45: Change “low physical activity” to “insufficient physical activity”.

ANSWER: Change has been made (see line 46).

  • Lines 45-47: Rewrite the following sentence:

“Low physical activity, smoking, leanness, low testosterone levels and decreased 25(OH)-vitamin D levels are sarcopenia developing factors in elders [4].”

ANSWER: Since smoking and leanness are not considered in intervention performed in our study, we have deleted these words resulting in a new sentence (see lines 46-48).

  • Provide a reference(s) to support the following sentence.

“Therefore, vitamin D deficiency influences the levels of certain minerals,”

ANSWER: Reference 6 has been cited (see line 51).

  • Lines 64-66: Rewrite the following sentence:

“Healthy elder women daily supplemented with magnesium oxide (MgO) for 12 weeks seem to improve physical performance, suggesting a role for Mg supplementation in preventing or delaying age-related decline in physical fitness [16].”

ANSWER: Sentence has been rewritten (see lines 65-68).

  • Lines 72-74: Provide at least two references to support the following sentence:

“Supplementation with NO precursors such as the amino acids L-arginine and L-citrulline or nitrate from beetroot extract, may help to control NO synthesis.”

ANSWER: References 22 and 23 have been cited. See line 80.

  • The study hypothesis needs to be added at the end of “Introduction”.

ANSWER: The hypothesis of the study is presented at the end of Introduction as suggested by the Reviewer (see lines 85-94).

Methods

  • “Materials and Methods” should be after the “Introduction” section.

ANSWER: Instructions for Authors indicate that Materials and Methods are in section 4.

  • Lines 261-262: What was the physical activity program? It needs to be explained.

ANSWER: A brief description of the physical activity program is indicated at lines 308-310.

  • Day-to-day test reliability, CV range, and intraclass correlation coefficients for the assessments need to be included for ALL the assessments.

ANSWER: For physical activity tests, this has been indicated in lines 335-337. For blood analysis and blood pressure, this is indicated in lines 353-355.

  • Add a schematic representation of the study procedures to the “Methods” section.

ANSWER: Schematic representation of the study is indicated in Figure 1.

Results

  • “Results” should be after the “Materials and Methods” section.

ANSWER: Instructions for Authors indicate that Results go after Introduction.

Discussion

  • “Discussion” should be after the “Results” section.

ANSWER: Instructions for Authors indicate that Discussion goes after Results.

  • The discussion section of this manuscript is the biggest concern. It needs to be elaborated and described more comprehensively to prepare for publication.

ANSWER: We have organized the Discussion in the following parts:

  • Physical activity is key to prevent sarcopenia (lines 172-191).
  • Metabolism activation to perform physical activity needs the presence of specific hormones such as testosterone. Low testosterone levels appear in sarcopenia (lines 192-217).
  • In addition, vitamin D is as well necessary for muscle function. This micronutrient presents low circulating levels in sarcopenia (lines 218-232).
  • The increase of vitamin D might favor an increase the levels of minerals, such as Ca and Mg. These minerals play a key role in muscle function and are instrumental for bone homeostasis (lines 233-259).
  • Once key hormones and micronutrients in muscle function and sarcopenia prevention have been presented, we proposed to increase their circulating levels by using NO precursors. We observed that only vitamin D increased, but not testosterone, Ca and Mg, suggesting that longer interventions are necessary to address this point (lines 260-269).

  • What is the novelty of this study? It needs to be mentioned in the “Discussion”.

ANSWER: The main novelty of this study is that one of the first circulating parameter that changes with supplementation is vitamin D, which deficiency correlates with sarcopenia development (see lines 274-278).

  • The Study limitations part is completely missing from the manuscript, and there are numerous limitations to be listed.

ANSWER: Limitations are indicated in lines 270-274.

Round 2

Reviewer 2 Report

Thank you for addressing the comments I suggested in order for your manuscript to be improved.

Indeed, the paper is more scientifically sound and complete now, given its limitations. However, those are appropriately presented.